# Athlete perceptions of virtual reality and barriers to its use in sport: A qualitative examination

**Jarad A. Lewellen** **\*, Eric Baker, Peter R., Giacobbi Jr**

School of Sport Sciences, West Virginia University, Morgantown, West Virginia, United States of America

\* jal00034@mix.wvu.edu

## Abstract

Though research has demonstrated the practical utility of virtual reality (VR) in settings such as the military, medicine, and counseling, VR in sport has become a research focus only in the last several years. As such, it is important to understand how this evolving technology is perceived and accepted by athletes. The purpose of this qualitative study was to examine VR use among athletes with experience using VR technology, uncover potential barriers to VR use in sport, and explore possible solutions to these barriers. Fourteen athletes with previous experience using VR headsets participated in semi-structured interviews exploring their experiences and perceptions of VR. The interviews were transcribed, cleaned, and coded in line with Braun and Clarke's qualitative framework. Using inductive and deductive analyses, we observed nine primary themes: where, when, why, what, frequency and duration, experience, acceptance, primary barriers, and overcoming barriers. Participants suggested a hesitancy among athletes and coaches to accept and adopt VR as a training tool. The primary barriers to adoption were cost, cybersickness, coach attitudes, awareness, understanding, and visibility. Applied implications focus on addressing and overcoming barriers by providing VR education and training to potential users and increasing accessibility to VR. Future research should expand to other populations (e.g., youth and professional athletes) and different versions of VR and extended reality technology (e.g., mixed reality), as well as examine effective ways to implement VR in applied settings. This study provides insight on athletes' perceptions of VR and contributes to the growing foundation of VR literature in sport.

## Introduction

Virtual reality (VR) is an evolving technology in which users can enter a virtual world and interact with their environment using a fully-immersive (e.g., a head-mounted display or a cave automatic virtual environment) or a semi-immersive system [1]. The use of VR has led to advances in various fields, and it has been referred to as the next stepping stone in technological innovation [2]. With its growing popularity, there

**Data availability statement:** DOI: 10.17605/OSF.IO/P4QMA All transcript files are available from the OSF repository at the link above. Data has been de-identified and the beginning of the interviews where identifiable demographics are discussed with the participant have been removed.

**Funding:** The author(s) received no specific funding for this work.

**Competing interests:** The authors have declared that no competing interests exist.

are broad potential applications for VR in numerous contexts. For instance, the military has begun using VR for various reasons, including landmine detection training [3] and managing post-traumatic stress disorder [4]. Within the medical field, VR has been used to explore and understand human anatomy [5] and for surgical simulation [6]. It has also been used in the treatment of phobias [7], in rehabilitation settings [8], and various other fields [2]. Thus, VR has promise for transferability and improved performance in a applied settings [9].

VR has also become an area of interest in sport settings with a range of experimental studies having been conducted. This technology allows athletes to immerse themselves in a virtual world and participate in various sport and exercise activities [10]. Early laboratory-based studies established the effectiveness of VR as a training tool using experimental designs. For example, Craig et al. [11] conducted two experiments with thirteen and eight non-soccer players, respectively, in which they used a virtual reality headset to simulate ball trajectory with and without sidespin. They then compared the performances to expert soccer players from a previous study [12]. Their analysis showed that ball-spin in a VR simulation led to large errors in perceptual judgment even in high-level soccer players, suggesting that virtual environments can replicate realistic real-life scenarios. While examining decision-making in rugby, Correia et al. [13] used a VR headset with head-tracking technology to examine 46 rugby players' perception-action coupling (i.e., a person's actions that result from their perception of relevant information) and showed that participants' decision-making was significantly influenced by their perception of emerging gaps in play and that VR could serve as a tool to facilitate that training.

In more recent years, researchers have made attempts to study the transferability of VR technology to real world sport settings by making simulations more realistic in a way that mimics sport-specific challenges. In one study examining the transferability of VR to real world behaviors, Harris et al. [14] compared expert and novice golfers in a virtual putting simulation to examine how performance was affected by training in VR. The results of this study indicated that skills can be learned in VR and transferred to the real-world, as evidenced by novice golfers showing comparable improvements in putting accuracy during real-world and VR practice. In another experiment examining transferability, Pagé et al. [15] showed that participants trained to perform basketball plays in VR outperformed the non-VR control group when performing transfer tasks, suggesting that that VR training may be transferable to real-world situations. While these results showed that VR could be used for performance enhancement in sports, this potential appears to be largely unmet with a lack of evidence that VR is used by athletes and sport psychology consultants in practical settings.

Diffusion of Innovations theory [16,17] and the Technology Acceptance Model [TAM; 18] explain the process of adoption and acceptance of technology, respectively, and can provide insight into the perceived lack of VR application in sport. Specifically, Diffusion of Innovations theory explains how and why certain discoveries, products, or technologies are adopted by some individuals early while others are late adopters. When adopted as part of a social system, in this case VR in sport,

technology has become diffuse, or it has spread. The model considers five categories of adopters: 1) innovators (i.e., those who want to be the first to try the innovation, 2) early adopters (i.e., those who want to be opinion leaders), 3) the early majority (i.e., those who are not leaders but want to adopt new products or technologies before the average person), 4) the late majority (i.e., those who are skeptical of change but will adopt a new product or technology when the majority has), and 5) laggards (i.e., those who are conservative and the hardest group to change). To date, it is not clear where athletes and coaches fit into the adoption categories of Diffusion of Innovations for VR.

Meanwhile, the TAM is based on the Theory of Reasoned Action [19] and the Theory of Planned Behavior [20], and it suggests that individuals are more willing to use a technology depending on their perception of its usefulness and how easy it is to use. As such, attitudes, social norms, and perceived behavioral control predict intentions to adopt a new technology and possible usage. The TAM model has been used extensively in other disciplines, particularly in education [21], marketing [22], and healthcare [23]. Further, it has been used to examine the acceptance of VR in domains such as pediatrics [24] and education [25]. While these studies have contributed to the understanding of VR acceptance, this line of research largely appears to have evaded research attention in sport. However, possibly the only study to examine acceptance of VR in sport using the TAM suggests that perceptions of usefulness, ease of use, enjoyment, and subjective norms all predicted intent to use VR [26].

One qualitative study examined professional European soccer coaches' perceptions of barriers to VR use [27]. The authors discovered four primary themes regarding barriers to VR use. These barriers included 1) practicality (e.g., concerns about time and space for proper use), 2) a lack of empirical evidence leading to skepticism about VR's value, 3) concern that the current quality of VR cannot replace practice and accurately replicate gameday environments, and 4) concern that VR might deliver too much data to players, thus leading to cognitive overload and subsequent performance decreases. While this information is critical to understanding barriers to VR use, it focuses on a specific population (i.e., professional European soccer coaches). As such, given the lack of understanding of VR application in sport and qualitative research on athlete perceptions of VR, it would be useful to examine the VR experiences of athletes from a variety of personal and athletic backgrounds.

To our knowledge, no research has qualitatively examined individual experiences through a contextual lens within this realm. Based on the current literature and the need to gain a deeper understanding of VR use in sport, the purpose of this qualitative study was to explore athletes' experiences with VR and their perceptions about its use in sport. More specifically, we conducted in-depth interviews to 1) examine VR use among athletes with experience using it, 2) uncover potential barriers to VR use in sport, and 3) explore possible solutions to these barriers.

## Methods

This study was conducted after approval was obtained from the West Virginia University Institutional Review Board (Protocol #2302726309). The informed consent process included steps to minimize possible coercion, and participants provided written consent before the interview was conducted. Data was subsequently de-identified and uploaded to the Open Science Framework repository. The study is reported using the Standards for Reporting Qualitative Research [SRQR; 28].

### Research design and philosophical orientation

This qualitative study explored athletes' views and experiences with VR. To our knowledge, this study is among the first of its kind which created a need for epistemological flexibility. As such, given its emphasis on practical experiences, the research aims, and consequences of inquiry, the study was guided by a pragmatic research philosophy [29] with focus on the practical experiences of athletes' use of VR in sport. Pragmatically oriented researchers are more concerned about addressing practical problems than the correspondence between theory, knowledge, and reality [30]. As such, we utilized a qualitative individual interview approach designed to develop a shared understanding of how athletes use VR, the major barriers and facilitators of its use, and ways to overcome the barriers.

## Researcher characteristics and reflexivity

The research team consisted of two doctoral students (JAL, EB) and one PhD qualitative researcher (PRG), all of whom are male. JAL and PRG had previously conducted qualitative research with athletes, and JAL had previous research and practical experience with VR. The research team engaged in active reflection and dialogue throughout the research process in order to minimize any biases or assumptions during data collection and analysis.

## Participants

The sample included 14 athletes aged 18–40 (M = 22; SD = 5.46) and consisted of 9 males and 5 females. Self-identified ethnicities consisted of White (*n* = 12), Hispanic (*n* = 1), and Asian (*n* = 1). Their average years competing in their sport was M = 11.06 (SD = 5.95) across various sports (with one participant competing in multiple sports). Participant sports consisted of golf (*n* = 7), softball (*n* = 2), rifle (*n* = 2), ice hockey (*n* = 1), tennis (*n* = 1), martial arts (*n* = 1) boxing (*n* = 1), weightlifting (*n* = 1), and shooting (*n* = 1). Participants' highest level of competition reached was either college (*n* = 13) or amateur (*n* = 1). On average, the interviews lasted 34.7 minutes. A detailed breakdown of participant demographics can be found in Table 1.

Participants were recruited via a purposive sample using publicly available NCAA coach emails, social media, and word of mouth. Inclusion criteria stipulated that participants were 1) at least 18 years old and 2) had used a VR head-mounted display (HMD) at least once. Of note, while some participants may have had previous experience with semi-immersive VR (e.g., a golfer using a golf simulator), interviews and subsequent coding focused primarily on their experiences using fully-immersive VR (i.e., an HMD).

## Data collection

Once an athlete agreed to participate, they were provided with a secure Zoom link for their scheduled day and time. The Zoom interviews were audio recorded and used to produce transcripts of each interview. All data was stored on a password-protected computer that was accessible only to the researchers directly involved with the study. Interviews were conducted between February 16, 2023, and February 13, 2024. The semi-structured interview guide used for this study

**Table 1. Self-reported participant demographics.**

| Participant ID | Age (Years) | Sex | Race | Primary Sport(s) | Competitive Level |
|---|---|---|---|---|---|
| Athlete 1 | 22 | M | White | Golf | College |
| Athlete 2 | 22 | M | White | Rifle | College |
| Athlete 3 | 25 | M | White | Ice Hockey | College |
| Athlete 4 | 19 | F | Hispanic | Softball | College |
| Athlete 5 | 19 | F | White | Softball | College |
| Athlete 6 | 19 | M | Asian | Rifle | College |
| Athlete 7 | 20 | M | White | Tennis | College |
| Athlete 8 | 18 | F | White | Golf | College |
| Athlete 9 | 21 | M | White | Golf | College |
| Athlete 10 | 21 | M | White | Golf | College |
| Athlete 11 | 40 | M | White | Martial Arts/Boxing Weightlifting/ Shooting | Amateur |
| Athlete 12 | 21 | F | White | Golf | College |
| Athlete 13 | 21 | M | White | Golf | College |
| Athlete 14 | 20 | F | White | Golf | College |

was partially based on questions developed by Munroe and colleagues [31] who explored the 4 Ws (i.e., where, when, why, and what) of athletes' use of mental imagery in sport. Interview questions for the present study included: "Where did your VR use take place?"; "How frequently have you used VR?"; "What skills do you believe VR can be used to train?"; and "What type of VR equipment have you used?" We also included questions, using the TAM [18], that were intended to gather information about participants' acceptance of VR as a training tool. Such questions included: "What was you experience like the first time you used VR?"; "Do you think VR is useful for your sport?"; and "Are you willing to use VR again in the future?" The interview guide is available as a supporting information file (S1 File).

## Data analysis

The transcribed interviews were examined for accuracy and edited for clarity as needed by re-listening to the interviews while reading the transcripts. The first author created a coding template which guided the analysis during the initial phases with adjustments made as the team worked together. All interviews were coded twice by independent team members. We used Braun and Clarke's [32,33] six phases of reflexive thematic analysis to analyze and present the data. First, the first author familiarized themselves with the data by transcribing, reading, and re-reading the interview transcripts and noting initial thoughts and impressions. Second, the first author coded all 14 transcripts, and initial codes were generated systematically across the entire data set. During the third phase, we sorted initial codes into possible themes and subthemes. At this stage, we used color-coded theme mapping to visualize the evolving themes. In phase four, we reviewed and refined the themes to ensure they followed a coherent pattern. More specifically, active reflection throughout data collection allowed the team to align quotes and label themes consistent with participants' views about VR in sport. Then, the second and third authors independently coded transcripts 1–7 and 8–14, respectively, to determine whether the themes aligned with the data set and to identify any additional codes that may have been missed in earlier coding stages. In Phase five, we further defined and refined the themes by examining the story that each theme told individually and as part of the overall data, identifying subthemes, and creating clear definitions and names for each theme. Finally, in phase six, we extracted data (e.g., quotes) that provided vivid and compelling examples of the themes. The reflexive thematic analysis allowed for epistemological flexibility in line with pragmatic thought [29]. This approach combined the use of inductive coding that reflected the content of the interview data with deductive analysis aligned with the 4 Ws [31] and TAM [18]. Finally, an audit was conducted with the data by an individual not affiliated with the study. This individual matched participant quotes with the themes with 100% accuracy.

## Results

We observed nine primary themes about athlete VR use: 1) where, 2) when, 3) why, 4) what, 5) frequency and duration, 6) experience, 7) acceptance, 8) primary barriers, and 9) overcoming barriers. Within each higher-order theme, lower-order themes provided additional detail about participant experiences.

## Where

Participants were asked the location in which they typically use or used VR. Their responses can be divided into two subcategories: personal space and facility. Their usage in a personal space primarily consisted of either their own home or the home of a friend. Athlete 3, an ice hockey goalie, described how he moved the furniture in his living room to create the space needed to replicate the movements associated with being a goalie. He stated, "I use it in just like a living room, kind of clear out enough space that I can move around and do all the stuff I need to do [as a goalie]." He also noted one use when he "actually went on the ice. I like, went on a lake with it, and did it one time on like a frozen pond…I just had my gear on and threw the headset on, too." Usage in a facility consisted of a team facility or a training facility. Athlete 5 described using VR "in the practice setting; we're just in our hitting facility." Meanwhile, Athlete 11 used VR at a boxing facility, explaining:

> You got into the actual boxing ring because it was a nice easy way to keep you from running into walls … they did somehow had it set up so that you were properly moving around the ring in the VR.

## When

Participants were asked when they typically use or used VR, and their responses were divided into two subthemes: personal time and designated time. In the athletes' personal time, they used VR for either recreation or deliberate practice. For example, Athlete 1 explained, "for me like it's more of just like a recreational thing where I'm just kind of doing it in my free time." Several participants also used it for deliberate practice, which was particularly useful when they were unable to train at their team facility. For example, Athlete 3, an ice hockey goalie, began using VR to train at home. He then found it particularly useful when the global COVID-19 pandemic arose: "when it was [peak] Covid and stuff, my goal was to [use VR] every day, and I did it pretty much every day."

The subtheme "designated time" refers to when the athletes used VR as part of an assigned or scheduled period. For example, Athlete 5, a softball player, spoke about using it during team practice: "So obviously there's other stuff going on [during practice], but it's just like a separate station, and we get to be there for about 5 to 10 minutes." Additionally, when asked how her team utilized VR before an upcoming competition, Athlete 4, another softball player, explained:

> So, tomorrow is usually just a hitting day before actual competition day. And through that, VR will be within the station. And that's kind of already known, just because that's very much in our protocol and our game plan and everything. So usually like before a game, the day before, virtual reality will be a station.

## Why

When asked what skills they have used VR to train, or that they could conceive as skills that VR could be used to train (i.e., *why* they have used or might use VR), the themes that we observed are categorized as their 'actual' reasons (i.e., examples of previous uses related to their sport) and their 'projected' reasons (i.e., ways they could envision VR use that they had not previously done or considered). The 'actual' reasons most participants had used VR for their sport focused on improving confidence and perceptual-cognitive skills. Athlete 5, for example, explained how her use of VR for softball practice (e.g., taking extra batting practice reps using the headset) before a game affected her confidence, "I would just say confidence is another big [reason], just feeling confident before you get into a real game scenario." The lower-order theme of perceptual-cognitive skills includes skills such as anticipation and decision-making, reaction time, and hand-eye coordination. For example, Athlete 3, an ice hockey goalie, used a hockey-specific app to practice skills needed to be a successful goalie. He stated:

> [While using VR], I'm working on rushing a cross [and switching from one side of the net to the other] after they make a pass, watching the puck travel and then watching it come in and reacting to it as I make the save … It was improving my hand-eye [coordination and] definitely improved my reflexes.

Several athletes also discussed 'projected' reasons for VR use, which included injury recovery and preparation. This was reflected by Athlete 3, who discussed how he thought VR might be useful if he ever got injured, especially playing a contact sport (i.e., hockey):

> There's no risk a player is going to actually hit you [in VR]. I think, like physical-wise, like your legs are still working when you're doing it. Your hands are still working. It would really depend on what the injury is, but I think it could definitely be useful in terms of like making sure there's still no contact.

Preparation was further divided into situational practice and imagery. Athlete 5 noted how she would consider using VR for softball in the future if the technology and apps continue to advance:

> You can kind of put yourself in a situation. So maybe like, act like there's ghost runners on the base and maybe be like, okay well, there's a runner on [second base]. Try and hit it to the opposite side so you can score.

For imagery, participants described how VR could be useful to visualize competitive environments before they happen in real life, or to be able to visualize performance when you are not able to play. For example, when discussing how he could have used VR as a form of preparatory imagery in which he could view or play different golf courses when he was unable to physically be there, Athlete 10 noted:

> [It would be useful] especially if you're in a cold climate. Like in high school, I didn't play golf December through February. Those 3 months, I never even swung the club. Imagine if I would have had the VR headset [to practice those courses]; I could be a little better.

**What**

This theme describes "what" type of VR the athletes have used and is further divided by the subthemes of device and application. All athletes used either a head-mounted display (HMD) such as a Meta Quest or a handheld headset used with a mobile device. Some participants also had experience with headsets that included attachments to use with their sports equipment. For example, Athlete 3 described being able to "put the headset on. You could attach the controllers to your gloves, and you'd use your head, you could hold your stick in your hand if you wanted." When asked what applications they had used on the headsets, athletes described applications by discussing its presentation and purpose. The presentation was either animated, such as when Athlete 10 explained, "they were all animated … like typical games that a teenager would play," or real (e.g., "The video you could tell had been taken on the actual [location]. So it was like familiar territory. And it's a walking camera." Athlete 11). The purpose of the application also varied. Some of the applications were created for general recreational use, such as when Athlete 11 described having "done a fair number of VR-type games, everything from Beat Saber to a couple of shooting [simulators], one of them more video game-type, one a more real type." Meanwhile, some athletes reported using sport-specific apps, most commonly for softball, golf, boxing, and ice hockey. For example, Athlete 3 used an "app called Sense Arena. It was a goalie-specific ice hockey training program where they were, you know, there were realistic drills and different scenarios." Another app, used by Athlete 4 and Athlete 5, was Win Reality, which is a baseball- and softball-specific app. Athlete 5 explained:

> [The app] will say how far you hit the ball. And if you swing it and miss, it'll say strike or ball. And then if your swing was too high or too low. So that's another reason I like it.

**Frequency and duration**

The frequency and duration of the participants' VR use varied greatly. While some athletes reported having used VR between one and five times, others have used it consistently for training. For example, Athlete 9 "only did [VR] once." In contrast, Athlete 3 used VR "pretty much every day" during his peak usage. Meanwhile, the duration of each use was categorized as either less than 10 minutes, between ten and thirty minutes, and more than thirty minutes. Athlete 4 described her typical uses as being "6-to-10 minutes, depending on how much time we have." For the intermittent duration (i.e., 10–30 minutes), Athlete 11 used it for "no more than 15 minutes." Fewer participants, typically the ones with more VR

experience, had sessions longer than thirty minutes, such as when Athlete 10 described their typical sessions as lasting "45 minutes to an hour."

## Experience

When participants were asked what their first experiences using VR for either sport or recreation was like, the primary themes we observed were enjoyment, immersion, and cybersickness. For example, Athlete 11 stated:

> I certainly had a feeling of amazement at the technology … being able to look around and see the place as if I was actually standing there as opposed to it being on a flat screen that I then pan-view. I mean that was pretty interesting.

Participants also described the enjoyment that came with using VR in a group setting (e.g., with teammates or friends). Athlete 13, for example, explained:

> It was a ton of fun. We put it [on] the TV, and chanting after someone misses one, or going on a hot streak, or something like that. But it was a cool experience, especially for the first-time use.

When discussing the feelings of immersion while using VR, we observed three subthemes: senses, realism, and physiological experiences. Most participants felt that the use of senses, particularly sight, sound, and touch, enhanced the experience, such as when Athlete 3 explained how sound impacted his sense of immersion: "whenever you got scored on, they do a little goal horn … Then there would also be cheers like, if you like, made a bunch of saves or anything like that." Noting the realism of the experience while playing a recreational football game, Athlete 1 explained:

> I was playing quarterback, and like I was just like looking around, like it looked like there was fans in the stands… just being able to be immersed in like the full condition, I guess. Like to see everything how it actually would be if you were actually in that position. It was just cool and just basically just how realistic it was.

Several participants were also struck by their body's physiological responses during use, such as when Athlete 8 described her experience using a roller coaster app:

> I would be sitting on the couch, or like in a chair or something, and it would feel like I was on the roller coaster. Like my stomach, I would get like the stomach drop feeling like when you go down, and then I'd kind of try to like go with [the roller coaster] myself to kind of like, really, really get that feeling … it did feel like you were on that roller coaster.

In fact, seven participants discussed experiencing symptoms that are analogous to cybersickness (i.e., experiencing symptoms of motion sickness without the physical motion). While the participants did not use the word "cybersickness" in their responses, the conversation about their symptoms emerged when they were asked to discuss how VR made them feel and what their experiences using it were like. Athlete 8, for example, noted "It would kind of just make me feel sick. My stomach, almost like, you know, you get a headache, and then you like, feel sick, really sick. That's how I felt." Athlete 11, when discussing the first time he used VR, stated, "it took a moment to kind of like, reorient … once that happened, I was fine for a little bit before I started getting the motion sickness." Of those who experienced cybersickness, the most common symptoms reported were headache, nausea, dizziness, and eye strain.

## Acceptance

Following the tenets of the Technology Acceptance Model (TAM), we asked questions about, and observed themes based on, participants' perceived ease of use, perceived usefulness, and their behavioral intentions (i.e., willingness) about

VR. Participants categorized whether VR was easy to use based on if, or how quickly, they acclimated to the headset. When probed with questions about the acclimation process when using the headsets, the subthemes that we observed were spatial awareness, lack of introduction, a learning curve, and cybersickness. Regarding spatial awareness, several athletes noted their difficulty in understanding where their body was in relation to their surroundings outside of VR. For example, Athlete 1 explained:

> You had to get kind of used to like where you were … just like spatial awareness, kind of at first was a little harder, because, like I remember, we were playing the dodging bullets game, and like I was like just like ducking around and like running into the couch. It was like you have to get used to it like spatially.

Several athletes also noted that they did not have any introduction to how to use the headset or what type of applications were available to use. As Athlete 8 stated, "I just used it. Like I didn't look up any things that like could happen, [or] how you should use it." Another commonality amongst participants was a learning curve for using VR. Many noted that, during their first use, they did not understand what they were doing or how to engage with the virtual environment. For example, Athlete 10 explained the following:

> Those first 10-15 min were like, how do I move? What do I do? Like, how do I play this game, or like whether it was a game, or like a simulation, or something like that. Of course I had to get used to like, I had no idea what I was doing.

In examining the participants' perceived usefulness of VR, their opinions appeared to hinge on whether they had used VR for their sport or for recreation. For example, Athlete 7, who had only used VR for recreation, felt it could be helpful "depending on the sport," suggesting that VR was more conducive to sports with "isolated motion" such as baseball and golf, where you do no not have to account for your surroundings. Meanwhile, Athlete 3, who used VR frequently to train skills as an ice hockey goalie, said it is "100% [helpful for ice hockey]. I think it's absolutely something that, it's a tool. I think it's a really useful tool to have."

Finally, participants' willingness to use VR in the future was sporadic. Several wanted to, or planned to, use it again in the future: "I would definitely be open to like trying it … technology is advancing every single day. So I would be down for that for sure" (Athlete 1). However, other participants said their future use was contingent on at least one of three factors: no cybersickness, more applicability, and more accessibility. As cybersickness is an unpleasant experience, several athletes noted that they would only use VR again if they were certain they would not feel sick. For example, when probed about how cybersickness impacted her willingness to use VR again, Athlete 12 said she "probably would want to use it more" if she did not experience cybersickness. Other athletes noted a perceived lack of application to their own sport, with Athlete 8 stating that she would "absolutely" use VR, specifically an HMD, if it was more applicable to her sport (i.e., golf) and that she "probably would [buy] one." The athletes that said they would use VR again if it was more accessible had most often only used VR that they did not personally own. For example, Athlete 9, who had only ever used VR at his friend's house, stated, "I've just never had like the opportunity again to use one. But if I was given the opportunity, yes, I would use it."

## Primary barriers

When asked what they perceived as barriers to more athletes using VR for training, participants reported cost, cybersickness, coach attitudes, awareness, understanding, and visibility as primary barriers. As Athlete 7 stated, "I think the biggest thing is still that cost, because that affects its accessibility. That affects who can use it." Some participants also felt that the possibility of experiencing cybersickness hindered desire to use VR. For example, Athlete 8 said that "being sick [after] using it for like 20 minutes wasn't worth the fun that I had for that 20 minutes … It's fun until you start feeling bad."

Several participants also noted a potential "generational gap" between coaches and athletes, and they felt that many coaches follow traditional practices and are less likely to adopt new technology to their coaching, particularly if they do not understand the technology. Athlete 1 elaborated:

> I feel like sometimes coaches are kind of more like, I guess, set in stone. … [T]hey can be like just focused on the past and what they've done, how they found success in the past rather than like changing their view on things and be willing to change and adopt new technologies and stuff.

Many athletes also discussed a general lack of awareness of VR's potential in sport; they perceive that most athletes who have not already used VR are not aware that it is an option for training. Athlete 4 explained:

> I would say they're not as aware as they should be. I feel like virtual reality right now is something that people see as a game…So I feel like right now, there's not a lot of awareness, I guess, in sports aspects, especially for athletes.

Several participants suggested that a lack of awareness may contribute to a lack of understanding (i.e., even if athletes are aware of VR, they do not understand how to use it for their training). For example, Athlete 5 stated:

> I think [athletes] know what VR is. I just don't think they know that it could benefit them in the long run. Actually, one of the baseball players here at [my college], he was like, 'well like, do you even get to like see real pitches?' And I was like, 'do you know what VR is?' …So I kind of had to tell him about it before he kind of just made conclusions on his own.

Several athletes also explained that VR is not widely marketed as a tool for sport training (i.e., poor visibility). For example, Athlete 10 explained, "it hasn't really been pushed by the market yet at all, and when it does, I'm sure it will blow up. But just the like inexperience of it, like you just don't see much of it [advertised for sport]."

### Overcoming barriers

When asked how barriers to VR use in sport could be addressed, suggestions included two subthemes: promoting education and increasing accessibility. Most participants felt that if athletes were educated on how and why they could or should use VR for sport, they would be more likely to adopt it as a training tool. Athlete 13 said:

> If I was able to go to a class or being able to [talk to] someone that is very knowledgeable about them, if I was able to know how to use it the right way and use it beneficially to myself, I feel like we're in tune to spending that money if it's gonna help me out in the long run.

We also observed multiple instances where the interviews evolved into discussions about the future of VR use in sport and ways it might eventually be effective, with the interviewer and participant co-developing ideas. These discussions also indirectly educated the participants on VR's potential uses. In one instance, Athlete 11 noted that he wanted to "change my answer" to a previous question based on his newly realized information. He had previously stated that VR could not be used for boxing because there was no haptic feedback when taking hits from opponents. As the interview evolved, however, he realized that VR may be used to train reaction time, which could be used for defensive training in boxing.

Furthermore, making VR more accessible might overcome barriers such as cost, visibility, and cybersickness. Athlete 13 further stated, "in a perfect world, obviously funding [would help]. If we had an endless budget, I think we'd probably have, between our two teams, 12 [headsets]." Athlete 11 also noted:

> As one or two big name people start to adopt it, a lot of that barrier will get broken. Like once the first mainline boxing coach starts using it and has a winning fighter, suddenly a lot of boxing coaches are going to get curious.

Notably, participants did not have any concrete suggestions for how to overcome cybersickness, but rather suggested that if that barrier could be overcome (e.g., with a newer version of VR that addressed underlying causes of cybersickness), it would make VR more accessible. As Athlete 8 stated, "My experience was bad, but if I was given the opportunity to use a newer, more advanced [headset], I would, because I think it's very fascinating."

## Discussion

The purpose of this study was to gain a deeper understanding of athletes' experiences and perceptions of VR use for sport, uncover barriers to its use, and explore potential solutions to those barriers. Guided by inductive and deductive analyses, we observed nine themes of athlete VR use: where, when, why, what, frequency and duration, experience, acceptance, primary barriers, and overcoming barriers. Though the research on VR use and its perception in sport is limited, our findings compliment previous research while uncovering new information on how athletes are using VR, its acceptance and adoption, and its trajectory in sport training.

In examining the themes of where, when, why, and what, the participants reported that they most often used VR for recreation during their personal time (e.g., while at home) and training during designated practice time (e.g., at a team practice facility). In fact, only Athlete 3 routinely used VR for training while at home, and that was due largely to the global COVID-19 pandemic when he was unable to travel to a team facility. Otherwise, it appears that athletes in this sample largely did not use VR to train for their sport on their own time. However, that may be largely attributed to our finding that several athletes did not possess their own VR equipment, but rather they only used it when a friend, relative, or team possessed it. Moreover, the applications used during personal time were most often recreational games, though many focused on sport or exercise (e.g., golf app, Beat Saber, etc.). Even so, VR use in this sample was more commonly used for recreation, rather than training, when used outside of a team environment. When used within the team environment, there appeared to be more structured use, such as using VR during a station in practice and using sport-specific apps for training (e.g., Win Reality).

It is also noteworthy that there was considerable variation in the frequency and duration of VR use among participants. While participants reported specific frequency and durations of use during designated time (e.g., 10-minute rotations at team practice), there was less structure in their use of VR during their personal time, with frequency and duration appearing to depend on external factors (e.g., whether they owned the equipment or how many other people wanted to use it).

When considering the five categories of Diffusion of Innovations theory [16,17], it appears that participants in this study believe most athletes are either the late majority (i.e., skeptical of change but willing adopt a new technology when the majority has) or laggards (i.e., conservative and the difficult to change), and their opinions about coach attitudes toward VR indicate that coaches may fall into the same two categories. This was evident in our finding that most athletes lack awareness and understanding about VR, but they may be more willing to use it if it is marketed toward sport and high-authority figures begin to promote its use. One of our participants even used the term "early adopters" when discussing this, as he felt that getting people who already use VR to train (i.e., early adopters) to talk about its effectiveness could encourage others to adopt it.

While research using the Technology Acceptance Model [TAM; 18] for VR is limited, particularly in sport, our findings support this growing area of research. For example, our participants' perceptions of how easy VR was to use and how useful it was appeared to impact their acceptance. Another major factor in their acceptance and willingness to use VR again in the future was cybersickness, with multiple participants expressing hesitation to use VR again on the chance they may experience cybersickness. This finding reflects those reported by Sagnier et al. [34], who found that intent to use VR was positively affected by perceived usefulness and negatively affected by cybersickness. Meanwhile, Mascret et al. [26]

found that perceptions of usefulness, ease of use, enjoyment, and subjective norms were all predictors of intent to use VR, which was consistent with our findings. Likewise, the athletes in the present study reported they enjoyed using VR in group settings (e.g., with friends or teammates) and believed that many coaches were unwilling to adopt VR as a training tool. These themes support the notion that subjective norms may play a large role in VR acceptance. As Mascret et al. stated, the more often athletes' significant others (i.e., parents, teammates, coaches, etc.) encourage them to use VR, the more they will find it useful, easy to use, and enjoyable, and the more likely they are to use it. Conversely, due to the influence of subjective norms, athletes may be less likely to use VR if those around them do not support or encourage its use. In short, subjective norms may benefit VR acceptance and promote intent to use it. However, subjective norms may also serve as a barrier. For example, if an individual is part of a group that holds negative opinions about VR or that has low understanding of how it can be used for sport, that individual may be impacted by the norms of that group and may therefore be less likely to accept it as a training tool.

Among the barriers we observed, cost and factors that affect buy-in (e.g., understanding and cybersickness) appeared to be the most significant. These findings are consistent with Greenhough and colleagues' [35] finding that major barriers to VR use include cost of the equipment, limited research within soccer, time available to use it, coach and support staff buy-in, player buy in, space to use the equipment, personnel to operate the equipment, and first impression bias (i.e., having a negative first experience and allowing that impression to affect their willingness to use VR again). Participants in the present study also noted that more "traditional" coaches may be less likely to adopt a new technology, which, in addition to the literature on social norms, agrees with Thatcher et al.'s [27] finding that coaches are notably more skeptical of VR technology when they have limited experience with it, and that some coaches characterized VR as "gimmicky."

Among their other results, Greenhough and colleagues [35] found that participants were more likely to use VR if influential clubs or influential others used it. This compliments our finding of a perceived lack of visibility of VR use from high-profile individuals. In addition, our participants reported that VR has not been widely marketed to athletes but instead is most often marketed as gaming platform. In fact, multiple participants noted their perception that many people do not understand the potential benefits of VR in sport because it is only seen as a game rather than a training tool. A survey on consumers' perception of VR potential conducted by Global Web Index [36] found that 64% of VR users were most excited about its potential in gaming, which further demonstrates that the general perception of VR does not appear to focus on sport training.

Another major barrier to VR use in sport is a lack of understanding about how to use the technology for training. While the participants identified that other athletes do not generally understand how to use VR, we observed that even several participants did not fully understand how VR can be used as a training tool in sport, even though they had previous exposure to it. In several instances, when participants were asked what else they believed VR could be useful for, they were only able to think of concrete examples that involved using VR while physically playing a sport, rather than more abstract examples like using VR for mental skills such as arousal regulation or imagery. For example, one participant noted that they did not believe VR could be useful to swimmers because you cannot wear a headset while in the water, and another participant felt that VR would be less useful for tennis because it is difficult to replicate lateral movement in VR. In these cases, it appears that some athletes, even those with prior VR exposure, were not aware of many applications and training possibilities.

## Applied implications

Though barriers persist, identifying ways to overcome them could be the first step toward broader VR use in sport, and our findings in this regard have several applied implications for sport psychology consultants, coaches, and athletes. We identified that two ways to overcome these barriers are to provide education and increase accessibility. Research on technology acceptance has demonstrated a positive relationship between education and training and technology adoption [37]. Thus, providing training to users before their first use may improve their understanding of the technology, which can

in turn increase their perceived ease of use, perceived usefulness, and acceptance. This idea was demonstrated in multiple interviews in the present study when the interview evolved into the co-development of ideas between the participant and interviewer. One participant even acknowledged that his understanding of VR and its potential had changed over the course of the interview due to newfound knowledge he did not previously have. These interactions serve as an example of how education and training may influence users' understanding and ultimate acceptance of VR in sport.

Consultants could also provide coaches and athletes with training sessions on VR that are structured to increase acceptance via increased perceived ease of use (e.g., guidance on how to use VR) and perceived usefulness (e.g., education on its uses). The guidance provided to athletes on how to use VR should take care to minimize the risk of cybersickness and allow the user to experience a variety of applications that could be useful for their training. Education should include information on practical ways that VR can be used for their training, both generally and for their specific sport. For example, coaches that are new to VR have shown an interest in its use for injury recovery [27], so an educational session with coaches or medical or athletic training personnel might include demonstrations on how VR can be useful in aiding athletes' recovery from a torn anterior cruciate ligament (ACL) based on the current literature supporting VR for that purpose [e.g., 38,39]. Meanwhile, education on sport-specific use might include baseball and softball players using Win Reality to improve their pitch selection and decision making.

Consultants should also explore other uses of VR in conjunction with mental skills, and they might even create trainings based on athlete-specific needs. For example, if a golfer wants to visualize a course before playing it, a consultant could pair VR with imagery by creating a walkthrough of the course using a 360-degree camera and placing the athlete on the course in the VR headset using the captured footage. While in the virtual environment, the golfer could practice imagery techniques in preparation for their upcoming round. One study even used VR to deliver a pre-competition 20-minute mindfulness meditation that facilitated focus during competition and improved performance [40]. Such VR mindfulness interventions could be particularly useful in closed-skill sports that require high attention such as shooting, archery, darts, or golf.

Making VR more accessible may also overcome several barriers. For example, finding ways to provide athletes with VR at little to no cost to them may be one of the most effective ways to promote its use in sport. For example, sport psychology consultants might consider investing in their own VR equipment to use with athletes in sessions. Additionally, coaches and athletic departments should consider adding VR to the growing list of technologies currently used in athletics, though this may be difficult for teams with less funding sources such as youth and amateur. It would also likely boost VR use in sport if its visibility increased, particularly in marketing and endorsements. Given that consumers are more likely to buy a product endorsed by a celebrity athlete if there is congruence with the product and credibility from the endorser [41], such as a professional golfer endorsing VR as a training tool for golf, a rise in celebrity athlete use and endorsements of VR as a training tool could substantially impact its growth in sport.

It is also important to find ways to decrease the likelihood of cybersickness. While participants in the present study did not have any suggestions on how to do this, researchers have examined ways to address these adverse reactions to VR. For example, Carnegie & Rhee [42] suggest adjusting settings to improve depth-of-field and decrease visual discomfort while wearing a VR headset. Sepich and colleagues [43] found that decreasing cognitive workload during the VR task reduces the likelihood of cybersickness, particularly among new users. Stanney and colleagues [44] suggested that limiting motion parallax cues (i.e., the speed at which objects in the user's field of vision move relative to their depth) to allow users to acclimate to the virtual experience could decrease symptoms of cybersickness. Implementing these suggestions may allow users to experience minimal VR cybersickness, which may increase their levels of acceptance.

## Limitations and future directions

While our results contribute to the growing body of literature on understanding VR use in sport, some limitations are present and warrant discussion. First, while we set out to have a diverse sample, which in many ways was successful,

all but one participant was a college athlete. While we still received valuable insight, future research would benefit from including other competitive levels such as youth or professional. This may be particularly useful because younger generations appear to display higher optimism and motivation toward using newer technologies [45], which could lead to different findings based on age group. Our study also only focused on the use of VR headsets. While this was done intentionally to limit the scope of the study, future research should be conducted to understand athlete attitudes and experiences toward more VR technologies, and even technologies along the entire extended reality spectrum (e.g., augmented and mixed reality). Future research should also explore potential differences in approaches to VR based on the different sports and positions that athletes play, as the skills and training required for one athlete may vary significantly from others depending on the demands of their sport and position.

Our findings also revealed cybersickness to be one of the primary deterrents to repeated VR use. While researchers have begun exploring ways to limit the possibility of cybersickness, future research should continue to examine how to decrease adverse experiences with VR and make it more accessible. Furthermore, while we propose that consultants educate and train athletes on how to use VR for sport, there is currently no research on effective ways to implement this approach. If we are to follow the scientist-practitioner model of consulting, future research should provide empirical support for how to introduce athletes to VR in a way that promotes acceptance. Finally, the sample was relatively small, and the qualitative methods do not necessarily transfer to other athletes, so future research should continue to gather information on athlete perceptions of VR use to contribute to the growing understanding of how best to apply this tool to sport.

## Conclusions

The results from this study provide a deeper understanding of VR use and adoption among athletes, and we identified several barriers to its broader use in athletics. Several of those barriers support the current research, and others were novel findings that will contribute to the growing foundation of literature. In identifying those barriers, we also discussed ways to overcome them by promoting acceptance through the Technology Acceptance Model [TAM; 18] both practically and through future research. By doing so, we can contribute to making VR a more accessible tool for athletes and performers.

## Supporting information

**S1 File. Interview guide.**
(DOCX)

## Acknowledgments

We would like to thank the participants who took the time to take part in our interviews.

## Author contributions

**Conceptualization:** Jarad A. Lewellen.

**Data curation:** Jarad A. Lewellen, Eric Baker, Peter R. Giacobbi, Jr.

**Formal analysis:** Jarad A. Lewellen, Eric Baker, Peter R. Giacobbi, Jr.

**Investigation:** Jarad A. Lewellen, Eric Baker, Peter R. Giacobbi, Jr.

**Methodology:** Jarad A. Lewellen.

**Project administration:** Jarad A. Lewellen.

**Supervision:** Jarad A. Lewellen.

**Writing – original draft:** Jarad A. Lewellen.

**Writing – review & editing:** Jarad A. Lewellen, Peter R. Giacobbi, Jr.

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
