## [Decision Letter · Decision Letter 0]

26 Dec 2024

PONE-D-24-43684Athlete perceptions of virtual reality and barriers to its use in sport: A qualitative examinationPLOS ONE

Dear Dr. Lewellen,

Thank you for submitting your manuscript to PLOS ONE. After careful consideration, we feel that it has merit but does not fully meet PLOS ONE’s publication criteria as it currently stands. Therefore, we invite you to submit a revised version of the manuscript that addresses the points raised during the review process.

Please find a copy of the feedback provided from an external reviewer and myself as editor. 

We look forward to receiving your revised manuscript.

Kind regards,

Nick Dobbin

Academic Editor

PLOS ONE

Additional Editor Comments:

Dear Dr Lewellen,

Thank you for your submission. Please find some suggested change on the manuscript.

Abstract

Line 25: Please check the sentence structure here (i.e., technology doesn’t demonstrate research). VR has been used in….

Line 26: Really? VR was used in 1998 within the sport from my knowledge.

Line 29: Can I suggest you go with “limiting factors” rather than “barriers”.

Line 33: These themes feel like they were arrived at in a more deductive manner with those below (e.g., cybersickness) potentially be inductively derived. Seems you’ve used a mixed inductive/deductive approach in my view.

Line 34: Remove “our” throughout.

Line 37-38; This is vague with no additional context. I suggest you remove this from the Abstract or provide more insight.

Line 39: to 42: This is vague and hardly reflect the data you found. Please ensure this reflects your study and is not generic.

Introduction

Some of the sections here fail to set the scene for the study being proposed. Consider combining and shortening the paragraphs demonstrate use of, and effectiveness of, VR in sport. Then, move on to why this is not being overly well-adopted with use of the proposed theoretic models.

Line 61: You can remove this sentence, it’s fairly vague and generic. Get to the point with “Craig et al…”.

Line 61-68: Can you provide a bit more context for these studies including study design, participants, setting etc. This will be then provide the context for the next point in the subsequence paragraph.

Line 86 to 88: Why does lack of use equal scepticism and hesitancy? Nothing above does suggest scepticism and hesitancy

Methods

Line 158: Was this not convenience sampling?

Line 174: Hence a deductive rather than solely inductive approach?

Line 187: Can you briefly discuss which form of reflective thematic analysis you used given this is a tool-box of different approaches.

Line 188: data not date.

Results

I would encourage greater use of data in the results section to enhance the credibility. With no interview guide and limited insight into the questions, often the context is missing within the results section which limits the transferability of the results section. Thus, please incorporated context (e.g., “when asked about…. Few/some/several/many/all participants… This is reflected by XXXX who….”).

Line 202: I am not convinced these emerged as you say. You set out to find this information specifically.

Line 230: Please check your own instruction in the brackets.

Line 283: This section needs some development. It’s just a list of statements with some extremely short quotes.

Line 294: Was amazement only discussed by two athletes? I think the framing of the quotes needs some development.

Line 322: How many people commented on this? Did this conversation emerge, or did you ask participants about this?

Line 372: See my point above about “limiting factors”.

Line 374: Coaches opinion or attitudes?

Line 409:

Discussion

Line 434: Please check your aim aligns with the overall approach. Did you aim to end up with the themes that subsequently emerged from the data?

Line 461: What do you mean by coaches here? This reads as though they were your participants.

Line 486: Can you build on this final point a bit more.

Reviewers' comments:

Reviewer's Responses to Questions

**Comments to the Author**

1. Is the manuscript technically sound, and do the data support the conclusions?

Reviewer #1: Partly

2. Has the statistical analysis been performed appropriately and rigorously? 

Reviewer #1: N/A

3. Have the authors made all data underlying the findings in their manuscript fully available?

Reviewer #1: No

4. Is the manuscript presented in an intelligible fashion and written in standard English?

Reviewer #1: Yes

5. Review Comments to the Author

Reviewer #1: An interesting area which will become a major research and discussion issue in the future no doubt, although the manuscript has merit it is riddled with inconsistencies and I would debate if the manuscript answers your question.

I think you need to make a clear distinction of how VR is used ie deliberate practice or as a recreational tool during the leisure time of the athletes. The frame of mind an athlete approaches the use of VR is important to its use.

L93 there is a word missing

L145 can you be specific in what you did to mitigate the previous experiences and subjectivity?

L156 Table

Does the use of VR need to be discussed across different sports eg A3 and A11 take part in much more open skilled sports than the other participants. I think it is important to state A3 is an ice hockey goalie in the table.

Analysis L183-199 nice that you follow Braun and Clarke but coming from a pragmatic stand point would member checking have been useful to see if the participants agreed with your analysis and increase trustworthiness.

L206 More information and context needed here, how did the ice hockey player use his living room, at this point we do not know he is a goalie, even so the space he may need needs explaining.

L212 in the table A11 is described as Martial arts yet the quote here is about boxing.

L222 A3 took to VR during a lockdown? Would he have taken to VR if he was not forced into it or would he have continued with practice the traditional way.

L244 A3 talks about ‘I'm working on rushing across after they make a pass’ is this across his living room? I do not know ice hockey terminology but it seems more explanation is needed here.

L235-263 More description of how VR was used for each participant is needed this section left me questioning this. Is this used as an imagery script e.g. PETTLEP or are the participants sitting observing this going on, or something different again?

L258 what is meant by visualisation, in line with my last comment.

L311 and L339 A1, golfer, these comments are about him being a ‘quarterback’ and ‘dodging bullets’ both of these do not involve golf. I am not sure how these quotes help support the use of VR in sport. I am not sure A1 makes a quote about golf.

L351-357 This section seems contradictory A7’s suggestion about ‘isolated motions’ sports it might help and A3 saying 100% useful ice hockey which is not isolated motions.

A3 and A11 seem to make up the majority of the quotes, this is only my impression so may not be correct.

There is definite potential in the manuscript but the writing needs to make clear the areas I have highlighted.

6. PLOS authors have the option to publish the peer review history of their article (what does this mean?). If published, this will include your full peer review and any attached files.

Reviewer #1: No

---

## [Author Response · Author response to Decision Letter 1]

9 Feb 2025

Manuscript PONE-D-24-25939

Response to Reviewers

Dear Dr. Dobbin,

Thank you for providing us the opportunity to submit a revised version of our manuscript “Athlete perceptions of virtual reality and barriers to its use in sport: A qualitative examination” for publication in PLOS ONE. We are grateful for the time and effort that you and the reviewer dedicated to providing us feedback on our manuscript, and we are thankful for the insights you have provided that have helped us strengthen our paper. We have incorporated most of the feedback provided. The changes are highlighted in the Revised Manuscript with track changes file. Additionally, please see our response to each reviewer comment below in blue. Please note that Microsoft Word has a glitch that changes the line numbers when Track Changes is set to “All Markup,” and that our line numbers below reflect the accurate continuous line numbers in the Manuscript file with no track changes, or when the track changes function is set to “Simple Markup” in the track changes file.

Best,

Jarad A. Lewellen

Author Response: Thank you for bringing this to our attention. We have edited the manuscript to ensure it follows the PLOS ONE style requirements.

Author Response: Thank you for the clarification. We have uploaded the de-identified data to the OSF repository and updated the Data Availability statement.

Author Response: Our ethics statement has been moved to the beginning of the Methods section.

Response to Reviewer #1:

Abstract

Line 25: Please check the sentence structure here (i.e., technology doesn’t demonstrate research). VR has been used in….

Author Response: As suggested, we restructured the sentence to read “Though research has demonstrated the practical utility of virtual reality (VR) in settings such as the military, medicine, and counseling, VR in sport has become a research focus only in the last several years.” In Lines 25-27.

Line 26: Really? VR was used in 1998 within the sport from my knowledge.

Author Response: We agree that the sentence implies VR in sport is completely new. While it has been researched in sport as early as the late 1980’s, it has only recently become a largely studied area, with published articles increasing exponentially over the last 10+ years. See re-written sentence above (Lines 25-27).

Line 29: Can I suggest you go with “limiting factors” rather than “barriers”.

Author Response: Thank you for the suggestion! We understand your rationale, but the literature on this subject frequently uses the term “barriers” (e.g., Greenhough et al., 2022; Thatcher et al., 2020), so we prefer to maintain that terminology.

Line 33: These themes feel like they were arrived at in a more deductive manner with those below (e.g., cybersickness) potentially be inductively derived. Seems you’ve used a mixed inductive/deductive approach in my view.

Author Response: This is a great point, and we should have made that clearer. We have included “Using inductive and deductive analysis, we observed…” in Line 33. We also replaced the word “emerged” with “we observed” throughout the document.

Line 34: Remove “our” throughout.

Author Response: Thank you for the suggestion. We have removed “our” throughout.

Line 37-38; This is vague with no additional context. I suggest you remove this from the Abstract or provide more insight.

Author Response: Thank you for the suggestion. We have removed it from the abstract.

Line 39: to 42: This is vague and hardly reflect the data you found. Please ensure this reflects your study and is not generic.

Author Response: Thank you for pointing that out. We have added more context with study-specific findings in Lines 39-42.

Introduction

Some of the sections here fail to set the scene for the study being proposed. Consider combining and shortening the paragraphs demonstrate use of, and effectiveness of, VR in sport. Then, move on to why this is not being overly well-adopted with use of the proposed theoretic models.

Author Response: Thank you for the feedback. We have combined and shortened the paragraphs about early lab-based studies and more recent studies on transferability, as well as added context for the lab-based studies (See response to comment on Lines 62-72 below).

Line 61: You can remove this sentence, it’s fairly vague and generic. Get to the point with “Craig et al…”.

Author Response: Thank you for the suggestion. We have removed this sentence.

Line 61-68: Can you provide a bit more context for these studies including study design, participants, setting etc. This will be then provide the context for the next point in the subsequence paragraph.

Author Response: Thank you for the feedback. We have added context about participants study design, and setting (e.g., noted these studies as laboratory-based studies that used experimental designs) in Lines 62-72.

Line 86 to 88: Why does lack of use equal scepticism and hesitancy? Nothing above does suggest scepticism and hesitancy

Author Response: Thank you for bringing this to our attention. Our word choice here was perhaps influenced by where the study was going rather than the literature we had discussed to that point. The sentence has been restructured to reflect an unbiased introduction to theories of adoption; Lines 86-88.

Methods

Line 158: Was this not convenience sampling?

Author Response: While we can understand the argument for convenience sampling, we feel our approach was purposive since we intentionally selected participants based on specific criteria (e.g., previous VR use; the second inclusion criteria listed).

Line 174: Hence a deductive rather than solely inductive approach?

Author Response: Thank you for pointing this out. We have revised the manuscript throughout to more accurately reflect the combination of inductive and deductive approaches as discussed by Braun and Clarke (2006; 2019).

Line 187: Can you briefly discuss which form of reflective thematic analysis you used given this is a toolbox of different approaches.

Author Response: Absolutely! As noted by Braun and Clarke (2006; 2019), reflexive thematic analysis involves several different approaches. In practice, these approaches are not fixed and can be used interchangeably. Therefore, we combined inductive and deductive analyses for our approach, and we have made this distinction clearer in the manuscript (e.g., Lines 201-204)

Line 188: data not date.

Author Response: Thank you for pointing this out! We have corrected the mistake.

Results

I would encourage greater use of data in the results section to enhance the credibility. With no interview guide and limited insight into the questions, often the context is missing within the results section which limits the transferability of the results section. Thus, please incorporated context (e.g., “when asked about…. Few/some/several/many/all participants… This is reflected by XXXX who….”).

Author Response: Thank you for the suggestion. We added context throughout the results section by incorporating questions and probes from the interview (e.g., “when asked how her team utilized VR before an upcoming competition, Athlete 4, a softball player, explained…”). Of note, we also moved the quote from Athlete 11 in “Primary Barriers” to the “Overcoming Barriers” section and replaced it with a quote from Athlete 10 about visibility. These quotes better reflect their new respective themes and sections. We also provided the interview guide as supporting information.

Line 202: I am not convinced these emerged as you say. You set out to find this information specifically.

Author Response: That is a great point. We have changed the wording to “we observed” so as not to suggest the themes emerged solely from inductive analysis (Line 208).

Line 230: Please check your own instruction in the brackets.

Author Response: Thank you for pointing this out. We have deleted the instruction and added the athlete’s sport to the quote’s introduction in Line 241.

Line 283: This section needs some development. It’s just a list of statements with some extremely short quotes.

Author Response: Thank you for the suggestion. However, we are unclear on what kind of development is needed. The findings in this section do not possess latent meaning, so we feel the current structure provides enough information about the data recorded about this theme.

Line 294: Was amazement only discussed by two athletes? I think the framing of the quotes needs some development.

Author Response: Thank you for the feedback. We had originally pulled the term “amazement” from the quote by Athlete 11, the only athlete who used that term explicitly, though the sentiment was echoed by several other participants. However, after revisiting it, we felt that “enjoyment” better represented the overall theme and was reflected more by the chosen quotes. As such, we changed the name of the theme to “enjoyment.” We also adjusted the wording in Line 326 to allow a better flow in the section.

Line 322: How many people commented on this? Did this conversation emerge, or did you ask participants about this?

Author Response: Thank you for the feedback. In Lines 349-353, we added that seven participants noted symptoms of cybersickness, that the conversation emerged from other prompts, and that they did not explicitly use the term cybersickness.

Line 372: See my point above about “limiting factors”.

Author Response: Thank you for the feedback. As stated above, we feel inclined to proceed with the term “barriers” as that is what is commonly used in the literature.

Line 374: Coaches opinion or attitudes?

Author Response: This is a good point; given that the data focuses more on how athletes believe coaches act as a result of their opinions toward VR, we changed the phrasing to “coach attitudes” throughout the manuscript.

Line 409:

Author Response: There was no comment here, but we noticed a typo (“us” instead of “use”) and the use of the word “emerged” again, so we assume the intended comment may have been about one or both of those. As such, the typo has been corrected (Line 408), and we removed “emerged” from the section.

Discussion

Line 434: Please check your aim aligns with the overall approach. Did you aim to end up with the themes that subsequently emerged from the data?

Author Response: Thank you for the feedback. We revised the manuscript to clarify that we used inductive and deductive analyses (e.g., Line 469), and therefore that we observed themes rather than suggesting the solely emerged from the data. We are confident this more clearly demonstrates that out aims align with our approach.

Line 461: What do you mean by coaches here? This reads as though they were your participants.

Author Response: Thank you for bringing this to our attention. We intended to convey that the athletes in the study indicated that coaches’ attitudes toward VR fall in line with Diffusion of Innovations. The sentence was edited to remove confusion, and the word “coaches” was removed from the following sentence (Lines 494-498).

Line 486: Can you build on this final point a bit more.

Author Response: Thank you for the suggestion. We built out this point by providing clearer context and an example of how subjective norms may serve as a barrier to acceptance and use (Lines 519-524).

Reviewer #1: An interesting area which will become a major research and discussion issue in the future no doubt, although the manuscript has merit it is riddled with inconsistencies and I would debate if the manuscript answers your question.

Author Response: Thank you! We agree that this area of research is growing rapidly and will become a major focus in the future. After accounting for your comments and making the necessary revisions, we are confident our manuscript answers our questions.

Response to Reviewer #2:

I think you need to make a clear distinction of how VR is used ie deliberate practice or as a recreational tool during the leisure time of the athletes. The frame of mind an athlete approaches the use of VR is important to its use.

Author Response: Thank you for the feedback. We added more context to the beginning of the “When” section to distinguish that athletes were using VR for either recreation or deliberate practice within their personal use (Lines 228-236).

L93 there is a word missing

Author Response: Thank you for pointing this out! We have added the word “to” to correct the sentence.

L145 can you be specific in what you did to mitigate the previous experiences and subjectivity?

Author Response: Absolutely! We added that “The research team engaged in active reflection and dialogue throughout the research process in order to minimize any biases or assumptions during data collection and analysis” (Lines 147-149).

L156 Table

Author Response: There was no reviewer note included here, but we did edit the table to be in line with PLOS ONE’s formatting guidelines.

Does the use of VR need to be discussed across different sports eg A3 and A11 take part in much more open skilled sports than the other participants. I think it is important to state A3 is an ice hockey goalie in the table.

Author Response: Thank you for the suggestion. We did not explicitly discuss the difference in uses across different open vs. close skilled sports because the aims of the study were to provide perceptions about the use of VR in general. However, we included the following in the future directions section of the manuscript: “Future research should also explore potential differences in approaches to VR based on the different sports

---

## [Editor Report · Decision Letter 1]

16 Feb 2025

Athlete perceptions of virtual reality and barriers to its use in sport: A qualitative examination

PONE-D-24-43684R1

Dear Jarad A. Lewellen, 

We’re pleased to inform you that your manuscript has been judged scientifically suitable for publication and will be formally accepted for publication once it meets all outstanding technical requirements.

Kind regards,

Nick Dobbin

Academic Editor

PLOS ONE

Additional Editor Comments (optional):

Thank you for carefully considering the constructive feedback provided by the reviewers and editor. I am pleased to inform you that your manuscript has been accepted for publication in PLOS ONE. We sincerely appreciate your valuable contribution to the field.

Reviewers' comments:

None.

---

## [Editor Report · Acceptance letter]

PONE-D-24-43684R1

PLOS ONE

Dear Dr. Lewellen,

I'm pleased to inform you that your manuscript has been deemed suitable for publication in PLOS ONE. Congratulations! Your manuscript is now being handed over to our production team.

Kind regards,

on behalf of

Dr. Nick Dobbin

Academic Editor

PLOS ONE